# Effect of perioperative lidocaine infusion on the subjective quality of recovery after surgery: Protocol for an updated systematic review and meta-analysis

Qianli Huang[1], Changhui Shao[2], Wei Wei[3]*, Shan Ou[2]

**1** Department of Anesthesiology, Chengdu Women's and Children's Central Hospital, School of Medicine, University of Electronic Science and Technology of China, Chengdu, Sichuan, China, **2** Department of Anesthesiology, Chengdu integrated TCM & Western Medicine Hospital, Chengdu, Sichuan, China, **3** Department of Anesthesiology, West China Hospital, Sichuan University, Chengdu, Sichuan, China

\* weiw@scu.edu.cn

## Abstract

### Background

Lidocaine is increasingly used for surgical patients requiring general anesthesia. However, its clinical benefits on postoperative recovery quality are not well established. Our main objective aims to summarize the evidence regarding the effectiveness of perioperative lidocaine infusion on postoperative subjective quality of recovery (QoR).

### Methods and analysis

This protocol will be conducted according to the Preferred Reporting Items for Systematic Reviews and Meta-Analyses Protocols (PRISMA-P) guideline. This systematic review will include randomized controlled trials (RCTs) from their inception until December 31st, 2024 with no language restrictions. The major databases including PubMed, Embase, and the Cochrane library will be comprehensively searched and supplemented by a hand searching reference lists of all included articles. Searches will involve studies assessing the efficacy of the perioperative lidocaine infusion for improving postoperative QoR, in comparison to placebo, or on treatment. The two authors will independently screen studies, extract study data and assess bias risk of the studies. The subjective QoR (QoR-15, QoR-40) on postoperative day 1–3 will be defined as primary outcome, whereas secondary outcomes will include morphine consumption, incidence of postoperative nausea and vomiting, time to first bowel movement, time to first flatus, and length of hospital stay. A meta-analysis will be performed using Review Manager 5.3 software. Sensitivity analyses, subgroup analysis and publication bias will also be conducted. The evidence quality of pooled results

**Data availability statement:** No datasets were generated or analysed during the current study. All relevant data from this study will be made available upon study completion.

**Funding:** The research work was funded by a grant from the National Natural Science Foundation of China (grant no. 81971772, received by Wei Wei). The funders had no role in study design, data collection and analysis, decision to publish, or preparation of the manuscript.

**Competing interests:** The authors have declared that no competing interests exist.

will be assessed by the Grading of Recommendations Assessment, Development, and Evaluation (GRADE) approach.

## Discussion

This review and meta-analysis is anticipated to provide the evidence for the role of intravenous lidocaine on the subjective quality of recovery after surgery. In addition, the findings from this review will help clinicians with developing effective and safe perioperative anesthetic management regimens for surgery patients.

## Study registration

PROSPERO registration number: CRD42024585866

---

## Introduction

Postoperative recovery is most often defined by traditional surgical outcome measures including hospital length of stay, complication rates, and mortality rates [1]. With the advances in peri-operative care, serious complications have been decreasing [2] and a prompt recovery is expected for most surgery patients. Traditional measures have been incomplete to reflect patients' recovery profiles [3]. On the other hand, the impact of patients' perspective on recovery has garnered increasing attention of clinicians [4,5]. Previous studies have shown the importance of the implementation of effective strategies to improve patient-reported outcome measures for optimizing the patient recovery after surgery [6–8]. Quality of recovery (QoR) scale, such as the QoR-15 and QoR-40 scale, are patient-reported outcome measures involving physical, psychological and social domains [9,10]. These scales provide various aspects of postoperative recovery evaluated by the patient themselves and reflect the patients' perspective of healthcare quality, beyond just traditional outcomes, are widely used to evaluate the efficacy of interventions characteristics [11–15].

Lidocaine, an amino-amide local anesthetic agent, has been extensively investigated for its anti-nociceptive, anti-hyperalgesic, and anti-inflammatory properties [16]. A number of studies demonstrated that perioperative lidocaine infusion was effective to decrease postoperative pain [17,18]. In addition to analgesic effects, previous reviews have shown the administration of perioperative lidocaine reduced the risk of postoperative nausea and vomiting (PONV), the time to bowel function recovery and length of hospital stay in patients after abdominal surgery [19,20]. Besides, there is evidence suggesting that intravenous lidocaine could attenuate the incidence of postoperative cognitive dysfunction [21]. Lidocaine may also have the potential to improve patients' postoperative recovery. There have been several randomized clinical trials (RCTs) have attempted to evaluate the impact of intravenous lidocaine on postoperative QoR. However, emerging literature on intravenous lidocaine and postoperative QoR has shown divergent results.

Given the significant clinical implications of lidocaine, a systematic review and meta-analysis to evaluate the impact of intravenous lidocaine and postoperative QoR

is needed. The current literature on intravenous lidocaine and postoperative QoR is conflicting, necessitating further investigation. Despite a previous meta-analysis [22]showed that the benefits of intravenous lidocaine in enhancing QoR in patients undergoing laparoscopic bariatric surgery. However, the limited surgical types and sample size included made it difficult to draw definitive conclusions about the benefits in enhancing postoperative recovery quality. Another recent meta-analysis [23] concluded the efficacy of intravenous lidocaine for enhancing global QoR score [mean difference (MD) = 9.65, 95% confidence interval (CI): 6.33 to 12.97; $p < 0.00001$; $I^2 = 97\%$]. Sub-group analyses indicated the variety of surgeries as a possible source of the high heterogeneity. Moreover, several new studies have been published did not support these findings [24,25]. Hence, the efficacy of intravenous lidocaine in enhancing postoperative QoR in various surgical categories remains uncertain.

To further investigate the beneficial effects of intravenous lidocaine in postoperative QoR after surgery, we plan to conduct this protocol of a systematic review and meta-analysis to summarize and update the changes of the effectiveness of intravenous lidocaine in improving postoperative QoR in various surgical categories.

## Objectives

The propose of this review and systematic review is to address the following questions:

1. To investigate the specific impact of perioperative lidocaine infusion on the subjective quality of recovery in various surgical categories.

2. To formulate the recommended dosage range for the perioperative use of lidocaine.

## Methods

### Protocol registration and reporting

The present protocol was registered in the International Prospective Register of Systematic Reviews (PROSPERO) (registration number: CRD42024585866). The study protocol follows the guideline of the Preferred Reporting Items for Systematic Reviews and Meta- analysis Protocols (PRISMA-P) statement. The PRISMA-P-checklist is shown in S1 File.

### Search strategy and information source

A comprehensive search of PubMed, Embase and the Cochrane library databases will be performed from their inceptions to 31th December 2024 to find relevant studies. There will be no language restrictions. We plan to update the literature search during the process of writing the review manuscript. The key terms for search will contain "lidocaine", "surgical procedures, operative", "quality of recovery score", and "randomized controlled trials". In addition, we will manually search the references from retrieved articles to further identify additional relevant articles meeting the eligibility criteria. The search strategy of the respective databases will be detailed in S2 File.

### Eligibility criteria

The eligibility criteria for selecting trials designed according to PICOS (population, intervention, comparison, outcome, and study design) framework [26]. Eligibility will not be restricted by language, publication date, or type of surgery. Studies according to the following criteria will be considered inclusion: Population: studies involving adults (as defined in the individual study) undergoing any type of surgery under general anesthesia; Intervention: intravenous lidocaine during perioperative period, the intravenous lidocaine infusion must have been started intraoperatively, prior to incision, and continued at least until the end of surgery; Comparison: placebo or no treatment; Outcomes: the primary outcome is postoperative QoR measured using the subjective QoR (QoR-15, QoR-40) on postoperative day 1–3, while secondary outcomes are morphine consumption, incidence of PONV, time to first bowel movement, time to first flatus, and length of hospital stay; Study design: only RCTs will be included. The included studies must report at least one of the outcomes of interest.

We will exclude studies if they meet any of the following criteria:1) lidocaine was administered through non-intravenous routes, for example intramuscular, intra-articular, intrathecal or epidural routes; 2) studies which not involved intravenous lidocaine or placebo prior to and during the surgical procedure; 3) duplicate articles; 4) any letter, review, abstracts, editorial, case report or animal studies; 5) full-text articles cannot be obtained.

## Study selection and data extraction

All the retrieved studies will be imported to EndNote citation manager. After the removal of duplicates, two independent (Q.H. and C.S.) reviewers will screen all the articles based on the eligibility criteria to determine the suitability of the articles. Reviewers will begin by screening titles and abstracts, followed by the full texts of the retrieved studies screening. Any disagreements in the process of study selection will be resolved through discussion or a third senior reviewer will be consulted when necessary. The screening process will be summarized in a PRISMA flow diagram [27] (as shown in Fig 1).

Two independent reviewers (Q.H. and C.S.) will meticulously conduct data extraction for the included studies by using a standardized data extraction format prepared using Microsoft Excel. The Excel spreadsheet mainly comprises the following data, include first author, the year of publication, country, mean age, sex, total sample size, study design, regimens of lidocaine infusion (infusion rate, concentration, and time), type of surgery, and outcome measures. Studies will be classified according to surgical categories, such as abdominal, thoracic, orthopedic, and other surgeries. Any uncertainty occur in the process of data extraction will be resolved through discussions or a third reviewer will be consulted when necessary.

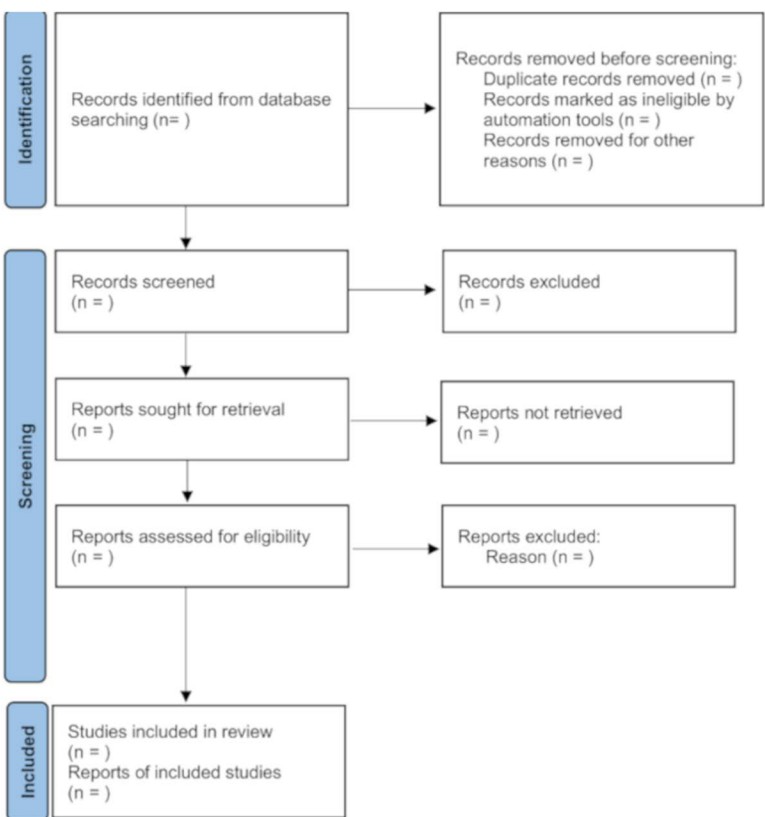

**Fig 1. PRISMA flow diagram of studies selection.**

## Dealing with missing data

In cases of crucial data is missing, we will contact the primary authors of the studies by email for relevant data upon reasonable request. If we receive no response in 14 days after the invitation, the selected studies will be excluded from the review.

## Risk of bias assessment

Two reviewers (Q.H. and C.S.) will independently examine risk of bias of the included trials using the Cochrane Risk of Bias (RoB-2) tool [28]. In detail, the tool is structured into for the following five domains: (1) the randomization process, (2) deviations from intended intervention, (3) lack of outcome data, (4) measurement of outcomes and (5) the choice of reported results. Each domain of ROB will be classified as low risk, high risk or some concerns. After that, the individual study's overall ROB will be determined. Any disagreements will be resolved through discussion or by involving a senior reviewer.

## Certainty of Evidence Assessment

We will assess the certainty of evidence using the Grading of Recommendations, Assessment, Development, and Evaluation (GRADE) approach [29], based on the following five parameters: risk of bias, indirectness, inconsistency, imprecision and publication bias. The certainty of evidence will be rated as high, moderate, low, or very low. Any discrepancies between investigators will be resolved by discussion or consultation with a senior reviewer, if necessary.

## Data synthesis

We will use Cochrane Review Manager (RevMan 5.3) to conduct data analysis. MD with 95% CI will be calculated for the continuous data and the risk ratio (RR) with 95% CIs will be calculated for the dichotomous data (e.g., PONV incidence), respectively. The heterogeneity will be evaluated by chi-square-based Q-test and quantified by $I^2$ statistic provided by Review Manager software. An $I^2$ value over 50% will be considered as substantial heterogeneity. Additionally, a p-value less than 0.1 will be considered statistically significant for heterogeneity. Assuming a heterogeneity across the studies, the random effects model will be carried out for outcome assessment, regardless of the heterogeneity. Subgroup analyses will be conducted based to explore potential sources of heterogeneity. Since a previous meta-analysis demonstrated a variation in the efficacy of intravenous lidocaine for QoR enhancement with the types of surgery [23], we will analyze the efficacy of lidocaine based on different subgroups of surgical categories (i.e., abdominal surgery, thoracic surgery, orthopedic surgery, and other surgeries) and anesthesia methods (i.e., total intravenous anesthesia vs. inhalation anesthesia). To identify other potential confounding factors, we will examine the impacts of age (i.e., < 60 years vs. > 60 years), sex (i.e., male predominance vs. female predominance), surgical time (i.e., < 120 min vs. > 120 min), and QoR scale (i.e., QoR-40 and QoR-15) on the QoR scores. Sex predominance will be defined as the proportion of a sex exceeding 50% among the recruited patients in a study. If substantial heterogeneity is detected, meta-regression will be considered to investigate the influence of specific study-level characteristics on the effect sizes. In addition, sensitivity analysis will be carried out to ensure the homogeneity of the results. When more than 10 RCTs are included in meta-analyses, the funnel plot asymmetry test and/or Egger's test will be used to explore potential publication bias [30]. If asymmetry in the funnel plot and a value of P < 0.05 according to Egger's test, reveals potential publication bias. All the statistical tests are two sided and P < 0.05 is considered statistical significance.

## Patient and public involvement

Patients and/or the public were not involved in the design, or conduct, or reporting, or dissemination plans of this research.

## Ethics and dissemination

The ethical approval is not applicable, as no individual patient data will be collected for systematic review. The results will be reported and disseminated through publication in a peer- reviewed journal.

## Discussion

QoR after surgery has been recognized as an important perioperative consideration in recent years. Currently, a growing body of literature has shown that perioperative intravenous lidocaine is effective in improving QoR after surgery. However, evidence was equivocal regarding the relationships between perioperative intravenous lidocaine with QoR. This review and meta-analysis will identify whether intravenous lidocaine interventions, might be effective in improving patients' QoR after surgery compared to no treatment, or placebo. The results from this review will provide clinical protocols regarding the use of lidocaine interventions to improve postoperative recovery quality.

This meta-analysis has several potential limitations. First, there is potential for heterogeneity. The variations in the timing and dosing regimens of lidocaine among the primary studies, may influence patient outcomes. Second, another potential limitation may include type of surgical procedures, the diversity of surgical procedures might impact the recovery profiles and pain experiences. Therefore, subgroup analysis or meta-regression will be considered to investigate the influence of potential confounders on the effect. Third, sample size from the included studies might be small, which will have an influence on the pooled results. Therefore, the application of GRADE criteria will contribute to a robust analysis. We expect to compile a comprehensive overview about the efficacy of lidocaine interventions in surgery. The findings will formulate the recommended dosage range for the perioperative use of lidocaine and a list of applicable surgical types.

## Supporting information

**S1 File. PRISMA-P 2015 Checklist.**
(DOCX)

**S2 File. Search strategies for included databases.**
(DOCX)

## Author contributions

**Conceptualization:** Qianli Huang, Changhui Shao.

**Data curation:** Qianli Huang, Changhui Shao.

**Formal analysis:** Wei Wei, Shan Ou.

**Methodology:** Qianli Huang, Changhui Shao, Wei Wei, Shan Ou.

**Software:** Qianli Huang, Changhui Shao.

**Writing – original draft:** Qianli Huang, Changhui Shao.

**Writing – review & editing:** Qianli Huang, Wei Wei, Shan Ou.

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
