## [Decision Letter · Decision Letter 0]

2 Feb 2025

PONE-D-24-41550Effect of perioperative lidocaine infusion on the subjective quality of recovery after surgery: protocol for an updated systematic review and meta-analysisPLOS ONE

Dear Dr. Wei,

Thank you for submitting your manuscript to PLOS ONE. After careful consideration, we feel that it has merit but does not fully meet PLOS ONE’s publication criteria as it currently stands. Therefore, we invite you to submit a revised version of the manuscript that addresses the points raised during the review process.

We look forward to receiving your revised manuscript.

Kind regards,

Stefano Turi

Academic Editor

PLOS ONE

Reviewers' comments:

Reviewer's Responses to Questions

**Comments to the Author**

1. Does the manuscript provide a valid rationale for the proposed study, with clearly identified and justified research questions?

Reviewer #1: Partly

Reviewer #2: Partly

2. Is the protocol technically sound and planned in a manner that will lead to a meaningful outcome and allow testing the stated hypotheses?

Reviewer #1: Yes

Reviewer #2: Partly

3. Is the methodology feasible and described in sufficient detail to allow the work to be replicable?

Reviewer #1: Yes

Reviewer #2: Yes

4. Have the authors described where all data underlying the findings will be made available when the study is complete?

Reviewer #1: Yes

Reviewer #2: Yes

5. Is the manuscript presented in an intelligible fashion and written in standard English?

Reviewer #1: Yes

Reviewer #2: Yes

6. Review Comments to the Author

You may also provide optional suggestions and comments to authors that they might find helpful in planning their study.

Reviewer #1: This protocol has two notable limitations that should be addressed. First, restricting inclusion to English-language publications introduces potential selection bias and may miss valuable evidence from non-English speaking regions, warranting removal of this restriction. Second, the protocol should better contextualize itself in relation to existing meta-analyses, particularly Hung et al. (PMID: 34547603), by clearly differentiating its scope, addressing identified knowledge gaps, and incorporating subgroup analyses to explore previously noted heterogeneity - these additions would strengthen the protocol's potential contribution to the evidence base while avoiding redundant investigation.

Reviewer #2: The overall structure of this manuscript is relatively complete, but there are still some parts that can be optimized, as follows.

1. Please supplement the detailed search strategies for each database.

2. In the sentence "In recent years, lidocaine, an amino-amide local anesthetic agent, have been extensively investigated for its anti-nociceptive...", the subject "lidocaine" is singular, so the verb should be "has" instead of "have".

3. Inclusion and exclusion criteria: In the inclusion criteria, the classification of surgical types only differentiates between elective or emergency surgeries and general anesthesia, without considering the impact of differences in surgical sites and trauma severity on the results, which may increase the heterogeneity of the study. It is recommended to further refine the criteria related to surgical types. For example, studies could be included by classifying surgical categories such as abdominal, thoracic, and orthopedic surgeries. In the exclusion criteria, the description of "lidocaine being part of an opioid - free anesthetic technique" is rather ambiguous. Different studies may have different definitions of opioid - free anesthetic techniques. It is suggested to clearly define this criterion or explain how to judge it during the actual screening process.

4. Regarding the subgroup analysis, only the assessment of the intervention effect based on patient characteristics and primary outcomes was mentioned, but the specific grouping factors were not clearly defined. It is recommended to list possible subgroup analysis factors such as age, gender, type of surgery, etc., and explain the basis for grouping and the expected analysis objectives.

5. Discussion section: When pointing out the potential limitations of the study, issues such as heterogeneity, diversity of surgical types, and small sample size were only briefly mentioned, without in - depth exploration of how to minimize these impacts in subsequent studies. It is recommended to propose specific countermeasures for each limitation. For example, in the study design stage, how to optimize the inclusion criteria to reduce heterogeneity, and how to expand the sample size, etc. The description of the expected research results is rather general, only mentioning providing suggestions for clinical practice and research. It is recommended to clarify the specific forms of expected results, such as formulating the recommended dosage range for the perioperative use of lidocaine and a list of applicable surgical types.

7. PLOS authors have the option to publish the peer review history of their article (what does this mean? ). If published, this will include your full peer review and any attached files.

**Do you want your identity to be public for this peer review?** For information about this choice, including consent withdrawal, please see our Privacy Policy .

Reviewer #1: No

Reviewer #2: No

---

## [Author Response · Author response to Decision Letter 1]

21 Feb 2025

Dear reviewer

We have special gratitude to you, for devoting your valuable time and energy to review our work entitled “Effect of perioperative lidocaine infusion on the subjective quality of recovery after surgery: protocol for an updated systematic review and meta-analysis” for giving constructive comments and valuable guidance. In line with this, the authors had exhaustively demonstrated and addressed questions and comments raised by reviewers using point-by-point responses as stated below.

Response: We have carefully checked throughout the manuscript.

Response: Thank you for your suggestion. We have included the ethics statement in the Methods section of our manuscript.

Response: Thank you for your suggestion. We have included a separate caption for each figure in our manuscript.

4. Please include captions for your Supporting Information files at the end of your manuscript, and update any in-text citations to match accordingly.

Response: Thank you for your suggestion. We have included captions for our Supporting Information files at the end of our manuscript and update any in-text citations to match accordingly.

Review Comments to the Author

Reviewer #1: This protocol has two notable limitations that should be addressed. First, restricting inclusion to English-language publications introduces potential selection bias and may miss valuable evidence from non-English speaking regions, warranting removal of this restriction. Second, the protocol should better contextualize itself in relation to existing meta-analyses, particularly Hung et al. (PMID: 34547603), by clearly differentiating its scope, addressing identified knowledge gaps, and incorporating subgroup analyses to explore previously noted heterogeneity - these additions would strengthen the protocol's potential contribution to the evidence base while avoiding redundant investigation.

Response: Thank you for your suggestion. First, we have removed the restriction of English-language publication. Second, we have added this discussion about the existing meta-analyses.

Reviewer #2:

1. Please supplement the detailed search strategies for each database.

Response: Thank you for your feedback. We have supplemented the detailed search strategies for each database.

2. In the sentence "In recent years, lidocaine, an amino-amide local anesthetic agent, have been extensively investigated for its anti-nociceptive...", the subject "lidocaine" is singular, so the verb should be "has" instead of "have".

Response: Thank you for pointing this out. We have rewritten this sentence as recommended.

3. Inclusion and exclusion criteria: In the inclusion criteria, the classification of surgical types only differentiates between elective or emergency surgeries and general anesthesia, without considering the impact of differences in surgical sites and trauma severity on the results, which may increase the heterogeneity of the study. It is recommended to further refine the criteria related to surgical types. For example, studies could be included by classifying surgical categories such as abdominal, thoracic, and orthopedic surgeries. In the exclusion criteria, the description of "lidocaine being part of an opioid - free anesthetic technique" is rather ambiguous. Different studies may have different definitions of opioid - free anesthetic techniques. It is suggested to clearly define this criterion or explain how to judge it during the actual screening process.

Response: Thank you for the insightful suggestion. First, we have revised the inclusion criteria as recommended. Second, we have modified the description of “lidocaine being part of an opioid - free anesthetic technique”, we appreciate your guidance.

4. Regarding the subgroup analysis, only the assessment of the intervention effect based on patient characteristics and primary outcomes was mentioned, but the specific grouping factors were not clearly defined. It is recommended to list possible subgroup analysis factors such as age, gender, type of surgery, etc., and explain the basis for grouping and the expected analysis objectives.

Response: Thank you for the insightful suggestion. We have added a detailed description of the subgroup analysis for the systematic review and meta-analysis.

5. Discussion section: When pointing out the potential limitations of the study, issues such as heterogeneity, diversity of surgical types, and small sample size were only briefly mentioned, without in - depth exploration of how to minimize these impacts in subsequent studies. It is recommended to propose specific countermeasures for each limitation. For example, in the study design stage, how to optimize the inclusion criteria to reduce heterogeneity, and how to expand the sample size, etc. The description of the expected research results is rather general, only mentioning providing suggestions for clinical practice and research. It is recommended to clarify the specific forms of expected results, such as formulating the recommended dosage range for the perioperative use of lidocaine and a list of applicable surgical types.

Response: Thank you for the detailed guidance. We have revised the discussion section to propose specific countermeasures for each limitation. We have also addressed potential limitations and how the findings will be interpreted and disseminated.

---

## [Decision Letter · Decision Letter 1]

17 Mar 2025

PONE-D-24-41550R1Effect of perioperative lidocaine infusion on the subjective quality of recovery after surgery: protocol for an updated systematic review and meta-analysisPLOS ONE

Dear Dr. Wei,

Thank you for submitting your manuscript to PLOS ONE. After careful consideration, we feel that it has merit but does not fully meet PLOS ONE’s publication criteria as it currently stands. Therefore, we invite you to submit a revised version of the manuscript that addresses the points raised during the review process.

We look forward to receiving your revised manuscript.

Kind regards,

Stefano Turi

Academic Editor

PLOS ONE

Journal Requirements:

Reviewers' comments:

Reviewer's Responses to Questions

**Comments to the Author**

1. Does the manuscript provide a valid rationale for the proposed study, with clearly identified and justified research questions?

Reviewer #2: Yes

2. Is the protocol technically sound and planned in a manner that will lead to a meaningful outcome and allow testing the stated hypotheses?

Reviewer #2: Yes

3. Is the methodology feasible and described in sufficient detail to allow the work to be replicable?

Reviewer #2: Yes

4. Have the authors described where all data underlying the findings will be made available when the study is complete?

Reviewer #2: Yes

5. Is the manuscript presented in an intelligible fashion and written in standard English?

Reviewer #2: Yes

6. Review Comments to the Author

You may also provide optional suggestions and comments to authors that they might find helpful in planning their study.

Reviewer #2: The article has been revised very well, and I highly appreciate the efforts made by the authors. However, there is still a minor issue that needs to be addressed.

Data Extraction and Analysis: The data extraction and bias risk assessment were carried out by two independent personnel, and the methodology is reasonable. In the data extraction form, an item of "specific infusion regimen of lidocaine (such as infusion rate and concentration)" can be added to enable a more comprehensive analysis of the impact of lidocaine use on the results. When conducting subgroup analysis, in addition to the factors mentioned in the article, the factor of different anesthesia methods can be considered for inclusion, because the anesthesia method may affect the efficacy of lidocaine.

7. PLOS authors have the option to publish the peer review history of their article (what does this mean? ). If published, this will include your full peer review and any attached files.

**Do you want your identity to be public for this peer review?** For information about this choice, including consent withdrawal, please see our Privacy Policy .

Reviewer #2: No

---

## [Author Response · Author response to Decision Letter 2]

19 Mar 2025

Dear Editors and Reviewers:

Thank you very much for giving us the opportunity to revise our manuscript (PONE-D-24-41550R1). We appreciate the helpful feedback from you and the reviewers. After carefully reading the comments, we have revised the manuscript point-by-point. Herewith we resubmit a revised manuscript for your assessment. Important changes are highlighted, and detailed responses to each comment are included below. We believe that these revisions have substantially improved the manuscript, and we thank you and the reviewers for their thoughtful comments, which are really helpful for the improvement.

Journal Requirements:

Response: Thank you! We have checked the reference list to ensure that it is complete and correct.

Reviewer #2: The article has been revised very well, and I highly appreciate the efforts made by the authors. However, there is still a minor issue that needs to be addressed.

Data Extraction and Analysis: The data extraction and bias risk assessment were carried out by two independent personnel, and the methodology is reasonable. In the data extraction form, an item of "specific infusion regimen of lidocaine (such as infusion rate and concentration)" can be added to enable a more comprehensive analysis of the impact of lidocaine use on the results. When conducting subgroup analysis, in addition to the factors mentioned in the article, the factor of different anesthesia methods can be considered for inclusion, because the anesthesia method may affect the efficacy of lidocaine.

Response: Thank you very much for your valuable comments. We have added the regimens of lidocaine infusion in the data extraction form and anesthesia methods in subgroup analysis.

---

## [Decision Letter · Decision Letter 2]

6 Apr 2025

Effect of perioperative lidocaine infusion on the subjective quality of recovery after surgery: protocol for an updated systematic review and meta-analysis

PONE-D-24-41550R2

Dear Dr. Wei,

We’re pleased to inform you that your manuscript has been judged scientifically suitable for publication and will be formally accepted for publication once it meets all outstanding technical requirements.

Kind regards,

Stefano Turi

Academic Editor

PLOS ONE

Reviewers' comments:

Reviewer's Responses to Questions

**Comments to the Author**

1. Does the manuscript provide a valid rationale for the proposed study, with clearly identified and justified research questions?

Reviewer #2: Yes

2. Is the protocol technically sound and planned in a manner that will lead to a meaningful outcome and allow testing the stated hypotheses?

Reviewer #2: Yes

3. Is the methodology feasible and described in sufficient detail to allow the work to be replicable?

Reviewer #2: Yes

4. Have the authors described where all data underlying the findings will be made available when the study is complete?

Reviewer #2: Yes

5. Is the manuscript presented in an intelligible fashion and written in standard English?

Reviewer #2: Yes

6. Review Comments to the Author

You may also provide optional suggestions and comments to authors that they might find helpful in planning their study.

Reviewer #2: The article is well revised, I appreciate the efforts made by the authors, and I recommend it for publication.

7. PLOS authors have the option to publish the peer review history of their article (what does this mean? ). If published, this will include your full peer review and any attached files.

**Do you want your identity to be public for this peer review?** For information about this choice, including consent withdrawal, please see our Privacy Policy .

Reviewer #2: **Yes: ** Xiongfeng Huang

---

## [Editor Report · Acceptance letter]

PONE-D-24-41550R2

PLOS ONE

Dear Dr. Wei,

I'm pleased to inform you that your manuscript has been deemed suitable for publication in PLOS ONE. Congratulations! Your manuscript is now being handed over to our production team.

Kind regards,

on behalf of

Dr. Stefano Turi

Academic Editor

PLOS ONE